# The Effect of Gut Bacteria on the Physiology of Red Palm Weevil, *Rhynchophorus ferrugineus* Olivier and Their Potential for the Control of This Pest

**DOI:** 10.3390/insects12070594

**Published:** 2021-06-30

**Authors:** Qian-Xia Liu, Zhi-Ping Su, Hui-Hui Liu, Sheng-Ping Lu, Bing Ma, Yue Zhao, You-Ming Hou, Zhang-Hong Shi

**Affiliations:** 1State Key Laboratory of Ecological Pest Control for Fujian and Taiwan Crops, Fuzhou 350002, China; lqx102250002046@163.com (Q.-X.L.); m17820288537@163.com (Z.-P.S.); lazyfeixu@163.com (H.-H.L.); lushengping2021@163.com (S.-P.L.); ma15534605636@163.com (B.M.); oaapgtx19900@163.com (Y.Z.); ymhou@fafu.edu.cn (Y.-M.H.); 2Fujian Provincial Key Laboratory of Insect Ecology, College of Plant Protection, Fujian Agriculture and Forestry University, Fuzhou 350002, China

**Keywords:** red palm weevil, gut bacteria, nutrient metabolism, intestinal immunity, pest control, paratransgenesis

## Abstract

**Simple Summary:**

Red palm weevil (RPW), *Rynchophorus ferrugineus* Olivier, is a destructive pest that often seriously infests palm plants. Because of the negative environmental effect and pesticide resistance caused by insecticide applications, it is important to develop novel green control strategies for this pest. The intestinal tracts of RPW are often colonized by multiple bacterial species that have promoting effects on the growth, development and immunity of RPW larvae. This review summarized the current understanding on the crosstalk between RPW larvae and their gut microbiota and pointed out the great potential of the development of microbial resource-based management methods for this pest.

**Abstract:**

Red Palm Weevil (RPW), *Rhynchophorus ferrugineus* Olivier, is a notorious pest, which infests palm trees and has caused great economic losses worldwide. At present, insecticide applications are still the main way to control this pest. However, pesticide resistance has been detected in the field populations of RPW. Thus, future management strategies based on the novel association biological control need be developed. Recent studies have shown that the intestinal tract of RPW is often colonized by multiple microbial species as mammals and model insects, and gut bacteria have been found to promote the growth, development and immune activity of RPW larvae by modulating nutrient metabolism. Furthermore, two peptidoglycan recognition proteins (PGRPs), PGRP-LB and PGRP-S1, can act as the negative regulators to modulate the intestinal immunity to maintain the homeostasis of gut bacteria in RPW larvae. Here, we summarized the current knowledge on the gut bacterial composition of RPW and their impact on the physiological traits of RPW larvae. In contrast with metazoans, it is much easier to make genetic engineered microbes to produce some active molecules against pests. From this perspective, because of the profound effects of gut bacteria on host phenotypes, it is promising to dissect the molecular mechanisms behind their effect on host physiology and facilitate the development of microbial resource-based management methods for pest control.

## 1. Introduction

Red Palm Weevil (RPW), *Rhynchophorus ferrugineus* Olivier, is a tremendously destructive pest for palm trees, including coconut *Coccus nucifera* L., date palm *Phoenix dactylifera* L. and coconut (*Cocos nucifera* L.) as well as urban palmscapes (e.g., those dominated by Canary Islands date palms (*Phoenix canariensis* Chabaud)) [1,2]. The female adults often lay eggs in wounds, cracks and crevices on the tree trunks, and then the larvae, the most damaging life stage, move into the interior of the palm trunk to construct tunnels and large cavities, with the production of the wet fermenting frass inside the galleries, and eventually lead to frond disfigurement and palm death [3,4,5]. Furthermore, RPW adults have the capacity to fly for a very long distance [6,7]. So far, RPW has been found in South Asia, the Middle East, the Mediterranean region [3,8], China [9], the Caribbean [10] and Malaysia [11], and has caused great economic loss, including expenses for removal of dead palms, lowered property values, degradation of recreational areas and urban wildlife habitats and potentially the harm and expenses associated with prophylactic pesticide applications to protect palms from weevils [12]. RPW has caused extensive economic damage in Egypt and the Gulf region [13,14], while the Mediterranean region has lost EUR 483 million [15]. In India, the removal costs for avenue *C.*
*nucifera* have been estimated to be USD 80–100 per tree (J.R. Faleiro pers. obs.) [12]. Pest severity is exacerbated because RPW feeding larvae are concealed, which makes early detection and control difficult [12]. Therefore, how to effectively control the infestation of RPW is still a major worldwide challenge at present.

Currently, exclusionary quarantines, pheromone traps and bio-control agents are employed to control RPW. However, insecticide applications are still the most effective way to protect palms against attack by palm weevils [12,16,17]. For example, these insecticides, containing organophosphates, carbamates, neonicotinoids and phenylpyrazoles, have been sprayed onto foliage, used as crown or soil drenches, used to dress wounds or injected into trunks or soil at the base of a trunk [12,18,19]. Unfortunately, resistance to synthetic insecticides such as chlorpyrifos, imidacloprid and lambda-cyhalothrin has been determined in the field populations of RPW [20]. Compared with the usage of chemical pesticides, biocontrol agents have the following advantages: low toxicity, target specificity and sustainability [21]. In this context, the employment of natural enemies is an important alternative to manage these notorious insect pests [9]. Two species of entomopathogenic pathogens, *Metarhizium anisopliae* [22] and *Beauveria bassiana* [23,24], the entomopathogenic nematode *Steinernema carpocapsae* [25,26] as well as several bacterial species such as *Pseudomonas aeruginosa*, *Serratia marcescens* and *Bacillus thuringiensis* (Bt) have been shown to be lethal to RPW insects [9,27]. For instance, laboratory bioassays revealed that RPW larvae exhibited lower boring activity, and a gradual decrease in feeding behavior following exposure to Bt [9]. However, the field trials performed so far on these biological agents have showed limited efficacy [12,28,29]. Therefore, new association biological control can be exploited for the development of novel management strategies [12]. Recently, it has been uncovered that the pathogenic fungus *Beauveria bassiana* can interact with the gut microbiota to promote the death of mosquitos [30]. The infection of Bt disrupted the gut integrity of *Spodoptera littoralis* Boisd, and then caused the translocation of gut bacteria, especially *Serratia* and *Clostridium*, into hemocoel to accelerate the death of this pest [31]. Consequently, these data suggest that the associated microbes of insects have the great potential to drive the development of novel microbes-based pest management strategies [16,32].

## 2. Gut Bacterial Compositions and Their Effects on the Physiology of Red Palm Weevil

As vertebrates, the gut of insects is usually colonized by diverse groups of microbes, which are generally known as gut microbiota, including bacteria, archaea, viruses, fungi and other protists. Among them, bacteria are the most predominant microbes, often referred to as gut bacteria [33,34,35]. It is difficult to generalize about gut bacteria of insects because of their vast ecological and taxonomic diversity [34]. The bacteria in the insect gut are usually Proteobacteria, Bacteroidetes, Firmicutes, Actinomycetes, Spirochetes and Verrucomicrobia [35,36,37]. Increasing evidence has found that insects have established mutual relationships with their intestinal bacteria [38]. For example, insects provide a stable survival environment and essential nutrients for gut bacteria, which in turn is involved in many physiological processes of insect hosts, including growth and development [39,40,41], metabolism [42], essential nutrient provisions [43], immunity and gut homeostasis [34,44,45]. Recently, gut microbiome has been found to play the essential role in nutrient allocation, mobilization and metabolism of *Nasonia vitripennis* Walker [46]. In *Dendrotonus ponderosae* Hopkins, *Dendrotonus valens* LeConte and *Ips grandicollis* Eichhoff, symbiotic gut bacteria can provide essential nutrients and be involved in digestion and detoxification of plant compounds [47]. Furthermore, increasing evidence supports that gut bacteria also influence the physiological fitness of the host through some specific metabolites [48]. Acetic acid, a metabolite produced by gut bacteria, can accelerate the growth rate of *Drosophila melanogaster* by upregulating the expression of insulin receptor and insulin-like peptide genes in the insulin/insulin-like growth factor signaling (IIS) [49,50]. Additionally, the gut bacterium, *Citrobacter* sp., can improve the resistance of *Bactrocera dorsalis* to the organophosphate insecticide, trichlorphon [51]. Consequently, gut bacteria profoundly affect the development, health and pesticide resistance of insect hosts.

Indeed, the intestine of RPW is also often inhabited by many bacterial species, showing important effects on host nutrition metabolism and development. The gut bacteria of RPW are mainly composed of Enterobacteriaceae, Lactobacillaceae, Streptococcaceae and Entomoplasmataceae. Some bacterial species, with the capacity to hydrolyze cellulose and hemicelluloses, can degrade polysaccharides to modulate the nutrition metabolism of RPW [42,52,53]. When the gut bacteria of RPW larvae were fully removed, the development rate from eggs to prepupa was dramatically prolonged, the body weight was reduced, and the contents of protein, glucose and triglyceride in the hemolymph were significantly decreased as well (Figure 1) [41,42]. These data suggest that gut bacteria can promote the growth and development of RPW larvae. In addition, several bacterial species can regulate the nutrient metabolism of RPW larvae. The sole introduction of *Lactococcus lacti* and *Enterobacter cloacae* to germfree RPW larvae can improve the content of protein, glucose and triglyceride in the hemolymph, respectively [41]. However, the molecular mechanisms underlying these processes are still unknown. Until now, it has been well defined that insect gut bacteria can be affected by the following factors, such as diet, environmental conditions and developmental stages. The disruption of gut microbiome homeostasis can dramatically impair the physiological fitness of some insect species [35,42,54,55,56,57]. For instance, the exposure of *Apis mellifera* L. to glyphosate and antibiotics can perturb its gut bacterial community structure, and then elevate their mortality upon the challenge of the opportunistic pathogen [58,59]. Recently, the ingestion of dsRNAs by the willow leaf beetle, *Plagiodera versicolora* Laicharting, results in dramatic alterations in the gut bacterial composition to accelerate its death [60]. Thus, these reports indicate it is promising to develop novel green pest control methods based on the disruption of gut bacterial homeostasis in insect pests [57].

## 3. The Role of Immune-Related Factors in Maintaining the Homeostasis of Gut Bacteria in Red Palm Weevil

The gut is the primary interface of material exchange between hosts and the external environment [61]. During feeding, a variety of bacteria can move into the gut with food, but not all of them are beneficial to the insects [35]. Therefore, the following question is very vital for insect health: how does the insect intestine discriminate between beneficial symbiotic and pathogenic bacteria quickly and accurately to achieve a sophisticated balance between tolerating symbiotic bacteria and eliminating the pathogens? Most gut bacteria release peptidoglycan (PGN), which is an important antigen for insects to recognize the invaded microbes and activate immunity [35,62,63]. It is well known that PGN is often recognized by peptidoglycan recognition proteins (PGRPs), the key immune receptors, to activate the immune responses of insects [64]. Previous investigations have identified several PGRPs in RPW, such as *Rf*PGRP-L2, *Rf*PGRP-L1, *Rf*PGRP-LB and *Rf*PGRP-S2, which are involved in the regulation of gut bacterial homeostasis in different ways [60,65,66,67]. For instance, *Rf*PGRP-L2 and *Rf*PGRP-L1 modulate the proliferation of intestinal bacteria by mediating the expression of antimicrobial peptide genes in gut epithelia cells [61,67]. Both *Rf*PGRP-LB and *Rf*PGRP-S1 act as negative regulators of RPW gut immunity to avoid the excessive activation of intestinal immunity by degrading PGN (Figure 2) [61,65,66,67]. However, whether these different PGRP members cooperate to modulate the homeostasis of gut bacteria needs further investigation. In addition, two innate immune signaling pathways were also found to perform different roles in the regulation of RPW gut bacteria homeostasis. Although both the *Rf*Relish-mediated-IMD-like pathway and *Rf*Spätzle–mediated-Toll-like pathway mediate the production of antimicrobial peptides to confer protection and maintain the homeostasis of gut microbiota, the IMD-like pathway plays the dominant role [68,69]. In sum, these data suggest that RPW has the tune mechanisms to maintain appropriate intestinal immune intensity to maintain the homeostasis of gut microbiota. 

Gut bacteria also mediate the resistance of their hosts to various pathogenic bacteria, parasites and other attacks [28]. Accumulating evidence showed that the axenic insects are more susceptible to pathogens and parasites in contrast with the conventionally reared ones [70,71]. Interestingly, the germfree RPW larvae exhibited impaired PO activity and a clearing ability to remove the invaded bacteria in hemocoel. It also died at a significantly higher rate upon facing the challenge of *Serratia marcescens*. Moreover, the transcript abundance of several pattern recognition receptors (PRRs) and antimicrobial peptides was significantly decreased in RPW germfree larvae [33]. Therefore, these data indicated that gut bacteria of RPW could enhance its immunocompetence by stimulating the expression of the important immunity-related genes, suggesting that gut bacteria have an immunostimulatory effect for their host and confer protection against pathogens.

## 4. The Promising Application of Gut Bacteria in the Pest Control

Insects are the most diverse and widely distributed group of animals on the earth. Many of them are pests or disease vectors, which has caused huge economic losses for agriculture and even posed serious threats to human health. Although insecticide applications have significantly dampened the occurrence of insect pests, the negative effects caused by chemical pesticides has forced humans to seek economic, efficient and sustainable pest management methods [28]. Therefore, impeding the infestation and spread of notorious insect pests efficiently and economically is still a major challenge [28,35,38]. Due to the profound effects of the associated microbes on insect hosts, Microbial Resource Management (MRM), which involves manipulating and exploiting microbiota for the management of insect-related problems, has been proposed [51,72]. Sterile insect technology (SIT) has been verified as an effective way to decrease the field population size of some insect pests by making and releasing sterile male individuals [35]. However, the mating competitiveness of the sterilized *Ceratitis capitate* Wiedemann males made by gama irradiation was significantly impaired [73]. Interestingly, it was found that the alterations in gut microbiota are the dominant reason for the impaired mating competitiveness of sterilized *C. captate* males. The introduction of gut bacterium, *Klebsiella oxytoca*, could rescue their mating competitiveness [74]. Therefore, this example suggests that it is feasible to solve the insect pest-associated problems by the manipulation of their gut bacteria.

It is much easier to complete genetic modifications in gut bacteria in contrast with insects. Consequently, genetic engineering has been employed to modify the symbiotic microbes to produce antipathogen effectors (termed paratransgenesis). That is, the dominant gut bacterial species can be modified by genetic engineering as the vector to produce some effective molecules to impair the physiology of insect pests [28,38]. For example, two symbiotic bacteria, *Pantoea agglomerans* and *Serrratia* bacterium strain AS1, have been successfully engineered by genetic modifications to secrete five different antimalarial peptides to disrupt the development of *Plasmodium falciparum* in the guts of *Anopheles* mosquitoes [75,76]. Moreover, termites can be killed by the introduction of a genetically engineered yeast with the capacity to produce a protozoacidal lytic peptide that can cause the death of protozoa in their gut [77]. Recently, gut bacteria have been demonstrated to show a synergistic role to accelerate the mortality of insect pests, which was caused by the pathogenic fungi [30], bacterium [78] and the ingestion of dsRNA [50]. These reports suggest that it is promising to deal with the insect pests-related problems by the manipulation and exploitation of their associated microbes. A central question for the usage of gut bacteria to reduce the infestation of insect pests is how to introduce the bacteria, and how to ensure their persistence in their field populations [79]. In RPW larva, two dominant gut bacterial species, *Lactococcus lacti* and *Enterobacter cloacae*, play major roles in the regulation of nutrition metabolism [41]. Interestingly, these two bacterial species have also been found in RPW larval frass and palm trunk tissue [80]. In addition, genetically engineered bacteria can be introduced into the RPW larvae through the bacterial inoculation into the healthy palm trunk. This evidence indicates that *L. lacti* and *E. cloacae* are promising candidates for paratransgenesis in this pest. Understanding of the diversity and functionality of the intestinal microbiome is the most important foundation for the exploitation of an MRM approach. Therefore, the deep dissection on the molecular mechanisms underlying the crosstalk between gut bacteria and their insect hosts will accelerate the development of gut bacteria-based pest management strategies.

## 5. Conclusions

As mammals, the insect’s intestine is often colonized by a community of commensal microbes, known as the gut microbiota. It has been well determined that the interactions between gut bacteria and their insect hosts profoundly affect the physiological fitness of insects, including reproduction, development, growth, immunity and behaviour. Furthermore, the maintaining of gut bacterial homeostasis is vital for insect health. In contrast with animal hosts, it is much easier to make the transgenic gut bacteria by genetic modifications. Thus, it is promising to develop the novel management strategies by genetically modifying the specific gut bacterium to dampen the occurrence of notorious insect pests. 

## Figures and Tables

**Figure 1 insects-12-00594-f001:**
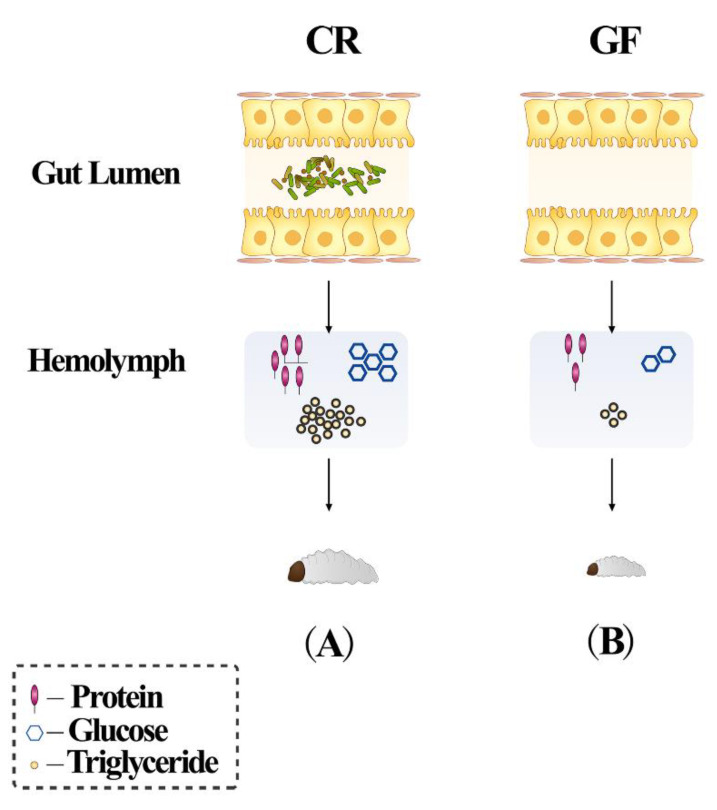
Gut bacteria promote the growth and development of RPW larvae via the modulation of nutrient metabolism. (**A**) The conventionally reared (CR) larvae have higher content of protein, glucose and triglyceride in the hemolymph; (**B**) the body weight, the content of protein, glucose and triglyceride in the hemolymph of germfree (GF) larvae were significantly decreased [41,42].

**Figure 2 insects-12-00594-f002:**
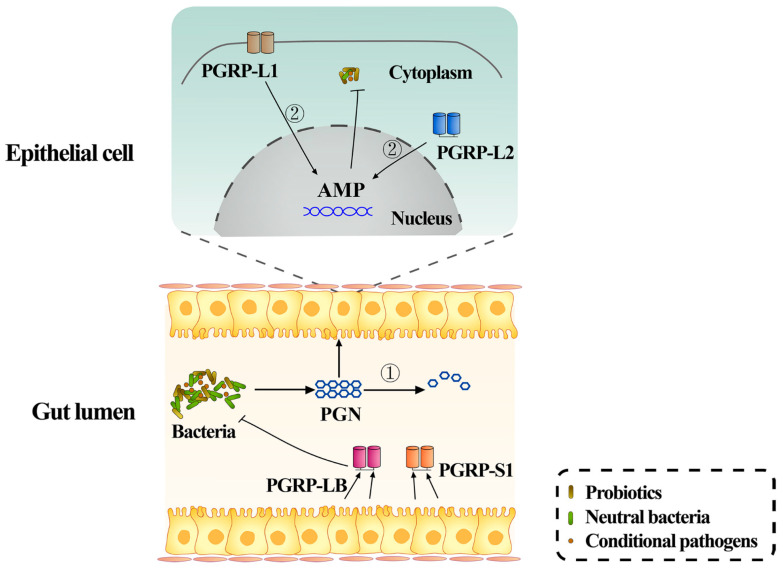
The role of RPW gut immunity in the regulation of intestinal bacterial homeostasis. Several PGRPs in RPW have been confirmed to be involved in the regulation of gut bacterial homeostasis via the different ways [61,65,66,67]. PGN: Peptidoglycan. AMP: Antimicrobial peptides. ①: Peptidoglycan degradation. ②: The activation of AMP synthesis.

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
