# Peer review of "The Effect of Gut Bacteria on the Physiology of Red Palm Weevil, Rhynchophorus ferrugineus Olivier and Their Potential for the Control of This Pest"

_insects, 2021, doi:10.3390/insects12070594_

Round 1

Reviewer 1 Report

These are my main comments on the review MS (Insects-1260771) entitled: "The effect of gut bacteria on the physiology of red palm weevil, Rhynchophorus ferrugineus Olivier and their potential in the control of this pest" by Liu et al.

This article is well motivated, yet quite short for a review paper, poorly structured, and misses on several key details.

The introduction and discussion provide no insight on how this MS relates to the various other ones cited in the text or concerns that have been raised by other researchers. This article should provide details on all these fronts to provide the proper context for the work. Authors do not present any hypotheses or expectations that could be connected to previous studies; adding these details will improve the paper. The authors should clearly explain WHY THE REVIEW WAS DONE, WHY IT WAS IMPORTANT, and HOW IT FITS WITH OTHER STUDIES. It should be clear and concise. The discussion should also include what outcome(s) they expect, and how it would help support or refute their hypotheses or answer their questions. Some of the authors statement would be much stronger if they tie their work to the body of literature that has built up on the invasion ecology and impacts of this pest in urban areas (see Journal of Pest Science, 2019 92(1), 143-156). The review lacks schemes to describe the phenology, distribution, palm infestation levels, management options for agricultural areas vs urban palmscapes, etc.

The discussion/conclusions lack real concluding remarks in my opinion, and if I was a practitioner, or if I was a consultant or extension scientist, I’d want to see these recommendations for my area or city. How applicable these findings are to the real world? The current version of this review is poorly structured and does not deal with the problem but rather only lists what has been done previously with gut microbiota. Adding this information would benefit the discussion. How would you address the issue of cryptic weevil habitats, high invasion capacity, low detection efficiency, and poor management organizations across different regions where these weevils have become problematic? What are the true benefits of gut microbiota to these programs?

Finally, my minor objections are about the language. I had difficulty understanding the English at times, which greatly affected my ability to understand whether the authors have entirely interpreted their connection to other studies appropriately. There are quite a lot of errors (awkward phrasing, poor vocabulary, syntax and grammar errors, etc.) throughout the manuscript.

Also, in some instances, the references are missing (e.g., line 47; I’d suggest adding here a reference Journal of Pest Science, 2019 92(1), 143-156) or are completely unrelated to the subject (e.g.: line 40); Overall, too little references are given for a very complex claims and a review paper in general.

The next draft of this paper will need to be dramatically different to have a chance at publication.

Wishing you best of luck.

Author Response

Dear Reviewer,

Thank you very much for your letter and advice on our manuscript. We have replied to your comments one by one, and revised and adjusted the content of the manuscript under the revision state according to the opinions. Specific replies are as follows:

1: The introduction and discussion provide no insight on how this MS relates to the various other ones cited in the text or concerns that have been raised by other researchers. This article should provide details on all these fronts to provide the proper context for the work. Authors do not present any hypotheses or expectations that could be connected to previous studies; adding these details will improve the paper. The authors should clearly explain WHY THE REVIEW WAS DONE, WHY IT WAS IMPORTANT, and HOW IT FITS WITH OTHER STUDIES. It should be clear and concise.

Response: Done. The mutual relationships between insects and their associated microbes are the important factors for their huge diversity. Although it has been well determined that gut bacteria have profound effects on insect physiology, less exploitations were conducted on these microbes from the perspective of pest control. Consequently, this is the major motivation for us to prepare this review. According to this comment, we have done the following revision: Currently, the exclusionary quarantines, pheromone traps, and bio-control agents are employed to control RPW. However, insecticide applications are still the most effective way for protecting palms from attack by palm weevils [12, 16, 17]. For example, these insecticides, containing organophosphates, carbamates, neonicotinoids, and phenylpyrazoles, have been sprayed onto foliage, used as crown or soil drenches, dressed wound, or injected into trunks or soil at the base of the trunk [12, 18, 19]. Unfortunately, the resistance to the synthetic insecticides, such as chlorpyrifos, imidacloprid and lambda-cyhalothrin, has been determined in the field populations of RPW [20]. Compared with the usage of chemical pesticides, biocontrol have the following advantages, such as low toxicity, target specificity, and sustainability [21]. So the employment of natural enemies is the important alternative to manage the notorious insect pests [9]. Two species of entomopathogenic pathogens, Metarhizium anisopliae [22] and Beauveria bassiana [23, 24], the entomopathogenic nematodes Steinernema carpocapsae [25, 26], and several bacterial species such as Pseudomonas aeruginosa, Serratia marcescens, and Bacillus thuringiensis (Bt) have been shown to be lethal to RPW insects [9, 27]. For instance, lab bioassays revealed that RPW larvae exhibited lower boring activity, and a gradual decrease in feeding behavior upon the challenge of Bt [9]. However, field trials on these biological agents performed so far have showed limited efficacy [12, 28, 29]. Therefore, new association biological control can be exploited for the development of novel management strategies [12]. Recently, it has been uncovered that the pathogenic fungus Beauveria bassiana can interact with the gut microbiota to promote the death of mosquito [30]. The infection of Bt disrupted the gut integrity of Spodoptera littoralis, and then caused the translocation of gut bacteria, especially Serratia and Clostridium, into hemocoel to accelerate the death of this pest [31]. Consequently, these data suggests that the associated microbes of insects have the great potential to drive the development of novel microbes-based pest management strategies [16, 32] (lines 61-86).

2: The discussion should also include what outcome(s) they expect, and how it would help support or refute their hypotheses or answer their questions.

Response: Done. In the final paragraph of our revised manuscript, we have presented a concrete suggestion on how to employ the gut bacteria to manage the infestation of RPW: A central question for the usage of gut bacteria to reduce the infestation of insect pests is how to introduce the bacteria, and to ensure their persistence in their field populations [79]. In RPW larva, two dominant gut bacterial species, Lactococcus lacti and Enterobacter cloacae, play the major roles in the regulation of nutrition metabolism [41]. Interestingly, these two bacterial species have also been found in RPW larval frass and palm trunk tissue [80]. And then the genetically engineered bacteria can be introduced into the RPW larvae through the bacterial inoculation into the palm trunk. So this evidence indicates that L. lacti and E. cloacae are promising candidates for paratransgenesis in this pest. The understanding on the diversity and functionality of intestinal microbiome is the important foundation for the exploitation in MRM approach. Therefore, the deep dissection on the molecular mechanisms underlying the crosstalk between gut bacteria and their insect hosts will accelerate the development of the gut bacteria-based pest management strategies (lines 217-229).

3: Some of the authors statement would be much stronger if they tie their work to the body of literature that has built up on the invasion ecology and impacts of this pest in urban areas (see Journal of Pest Science, 2019 92(1), 143-156). The review lacks schemes to describe the phenology, distribution, palm infestation levels, management options for agricultural areas vs urban palmscapes, etc.

Response: Done. We have done the following revision: Red Palm Weevil (RPW), Rhynchophorus ferrugineus Olivier, is a tremendously destructive pest for palm trees, including coconut Coccus nucifera L, date palm Phoenix dactylifera and coconut (Cocos nucifera)] to urban palmscapes [e.g., those dominated by Canary Islands date palms (P. canariensis)] [1,2]. The female adults often lay eggs in wounds, cracks, and crevices on the trunks, and then the larvae, the most damaging life stage, move into the interior of the palm trunk to construct tunnels and large cavities, with the production of the wet fermenting frass inside the galleries, and eventually lead to frond disfigurement and palm death [3, 4, 5]. Furthermore, RPW adults have the capacity to fly for a very long distance [6, 7]. So far, it has been found in South Asia, the Middle East, the Mediterranean region [3, 8], China [9], Caribbean [10], and Malaysia [11], and has caused great economic loss, including expenses for removal of dead palms, lowered property values, degradation of recreational areas and urban wildlife habitat, and potentially, expenses prophylactic pesticide applications to protect palms from weevils [12]. RPW has caused economic damage of 30% in Egypt and Gulf region [13, 14], while the Mediterranean region has lost 483 million Euros [15]. In India, the removal costs for avenue C. nucifera have been estimated to be $80–100 (US) per tree (J.R. Faleiro pers. obs.)[12]. Pest severity is exacerbated because RPW feeding larvae are concealed which makes early detection and control difficult [12]. Therefore, how to effectively control the infestation of RPW is still a great worldwide challenge at present (lines 42-60).

4: The discussion/conclusions lack real concluding remarks in my opinion, and if I was a practitioner, or if I was a consultant or extension scientist, I’d want to see these recommendations for my area or city. How applicable these findings are to the real world? The current version of this review is poorly structured and does not deal with the problem but rather only lists what has been done previously with gut microbiota. Adding this information would benefit the discussion. How would you address the issue of cryptic weevil habitats, high invasion capacity, low detection efficiency, and poor management organizations across different regions where these weevils have become problematic? What are the true benefits of gut microbiota to these programs?

Response: Done. We have done the following revision: In RPW larva, two dominant gut bacterial species, Lactococcus lacti and Enterobacter cloacae, play the major roles in the regulation of nutrition metabolism [41]. Interestingly, these two bacterial species have also been found in RPW larval frass and palm trunk tissue [80]. And then the genetically engineering bacteria can be introduced into the RPW larvae through the bacterial inoculation into the palm trunk. So this evidence indicates that L. lacti and E. cloacae are promising candidates for paratransgenesis in this pest (lines 219-225).

5: Finally, my minor objections are about the language. I had difficulty understanding the English at times, which greatly affected my ability to understand whether the authors have entirely interpreted their connection to other studies appropriately. There are quite a lot of errors (awkward phrasing, poor vocabulary, syntax and grammar errors, etc.) throughout the manuscript.

Response: Done. We have deleted the unscientific words, such as “Always,” “solidly proved,” and “absolutely” in the revised version. The other language issues have been fixed intensively, and the revisions on language have been highlighted in the manuscript.

6: In some instances, the references are missing (e.g., line 47; I’d suggest adding here a reference Journal of Pest Science, 2019 92(1), 143-156) or are completely unrelated to the subject (e.g.: line 40);

Response: Done. We have added the relevant content of this reference in lines 51-54, lines 56-59, lines 61-66, and line 77-80, respectively. As follows: and has caused great economic loss, including expenses for removal of dead palms, lowered property values, degradation of recreational areas and urban wildlife habitat, and potentially, expenses prophylactic pesticide applications to protect palms from weevils [12] (lines 51-54). In India, the removal costs for avenue C.nucifera have been estimated to be $80–100 (US) per tree (J.R. Faleiro pers. obs.)[12]. Pest severity is exacerbated because RPW feeding larvae are concealed which makes early detection and control difficult [12] (lines 56-59). Currently, the exclusionary quarantines, pheromone traps, and bio-control agents are employed to control RPW. However, insecticide applications are still the most effective way for protecting palms from attack by palm weevils [12, 16, 17]. For example, these insecticides, containing organophosphates, carbamates, neonicotinoids, and phenylpyrazoles, have been sprayed onto foliage, used as crown or soil drenches, dressed wound, or injected into trunks or soil at the base of the trunk [12, 18, 19] (lines 61-66). However, field trials on these biological agents performed so far have showed limited efficacy [12, 28, 29]. Therefore, new association biological control can be exploited for the development of future management strategies [12] (lines 77-80).

7: Overall, too little references are given for a very complex claims and a review paper in general.

Response: Done. We have added the following 28 references:

1     Wattanpongsiri, A. A revision of the genera Rhynchophorus and Dynamis (Coleoptera: Curculionidae). PhD dissertation. Oregon State University. 1966.

2     Murphy, S,T.; Briscoe, B. R. The red palm weevil as an alien invasive: biology and the prospects for biological control as a component of IPM. Biocontrol News Inf. 1999, 20, 35–46.

6     Hou, Y. M.; Wu, Z. J. Wang, C. F. The status and harm of invasive insects in Fujian, China, in Biological invasions: problems and countermeasures, ed. by Xie LH, You MS and Hou YM, Science Press, Beijing, 2011,pp. 111-114.

7     Han, Z.; Zhou, J.; Zhong, F.; Huang, Q. L. Research progress on damage and control of Rhynchophorus ferrugineus. Guangdong Agr Sci.2013, 40, 68-71

8     Llacer, E.; Negre, M.; Jacas, J. A. Evaluation of an oil dispersion formulation of imidacloprid as a drench against Rhynchophorus ferrugineus (Coleoptera, Curculionidae) in young palm trees. Pest. Manag. Sci. 2012, 68, 878-882.

10    Fiaboe, K.K.M.; Peterson, A.T.; Kairo, M.T.K.; Roda, A. L. Predicting the potential worldwide distribution of the red palm weevil Rhynchophorus ferrugineus (Olivier) (Coleoptera: Curculionidae) using ecological niche modeling. Fla Entomol. 2012, 95, 659–673.

11    Fong, J.; Siti, N.; Wahizatul, A. A. Virulence evaluation of entomopathogenic fungi against the red palm weevil, Rhynchophorus ferrugineus (Coleoptera: Dryopthoridae). Malays. Appl. Biol. J. 2018, 47, 25–30.

12    Milosavljevic, I.; El-Shafie, H. A. F.; Faleiro, J. R.; Hoddle, C. D.; Lewis, M.; Hoddle, M. S., Palmageddon: the wasting of ornamental palms by invasive palm weevils, Rhynchophorus spp. J. Pest. Sci. 2019, 92, 143-156.

13    El-Juhany, L.I. Degradation of date palm trees and date production in Arab countries: causes and potential rehabilitation. Aust. J. Basic Appl. Sci.2020, 4, 3998–4010.

14    El-Sabea, A.M.; Faleiro, J.; Abo-El-Saad, M.M.; The threat of red palm weevil Rhynchophorus ferrugineus to date plantations of the Gulf region in the Middle-East: an economic perspective. Outlooks Pest Manag. 2009, 20, 131–134.

15    Aldobai, S.; Ferry, M. Proposed multidisciplinary and multi-regional strategy for the management of red palm weevil. Scientific Consultation and High-Level Meeting on Red Palm Weevil Management”. organized by FAO and CIHEAM, Italy, 2017, pp. 29–31.

19    Hoddle, M.S.; Hoddle, C. D. How far can the palm weevil, Rhynchophorus vulneratus (Coleoptera: Curculionidae), fly? J Econ Entomol, 2016, 109,629–636.

20    Ahmed, R.; Freed, S. Biochemical resistance mechanisms against chlorpyrifos, imidacloprid and lambda-cyhalothrin in Rhynchophorus ferrugineus (Olivier) (Coleoptera: Curculionidae). Crop Prot. 2021, 143, 8.

21    Roh, J. Y.; Choi, J. Y.; Li, M. S.; Jin, B. R.; Je, Y. H. Bacillus thuringiensis as a specific, safe, and effective tool for insect pest control. J Microbiol Biotechn 2007, 17, 547-559.

22    Zhang, J.; Qin, W, Q.; Yan, W.; Peng, Z. Q. Detection of pathogenicity of Meatarhiziums against Rhynchophorus ferrugineus in laboratory. Chinese J Tropical C. 2012, 33, 899-905.

23    Hussain, A.; Rizwan-ul-Haq. M.; Al-Ayedh, H.; Ahmed, S. Al-Jabr, A.M. Effect of Beauveria bassiana infection on the feeding performance and antioxidant defence of red palm weevil, Rhynchophorus ferrugineus. BioControl. 2015, 60, 849-859.

24    Jalinas, J.; Güerri-Agulló, B.; Mankin, R. W.; López-Follana, R.; Lopez-Llorca, L. V. Acoustic Assessment of Beauveria bassiana (Hypocreales: Clavicipitaceae) Effects on Rhynchophorus ferrugineus (Coleoptera: Dryophthoridae) larval activity and mortality. J. Econ. Entomol. 2015, 108, 444-453

25    Llácer, E.; Martínez de Altube M. M.; Jacas, J.A. Evaluation of the efficacy of Steinernema carpocapsae in a chitosan formulation against the red palm weevil, Rhynchophorus ferrugineus, in Phoenix canariensis. BioControl, 2009, 54, 559-565.

26    Mastore, M.; Arizza, V.; Manachini. B.; and Brivio. M.F. Modulation of immune responses of Rhynchophorus ferrugineus (Insecta: Coleoptera) induced by the entomopathogenic nematode Steinernema carpocapsae (Nematoda: Rhabditida). Insect Sci. 2015, 22, 748-760.

27    Banerjee, A.; Dangar, T. K. Pseudomonas aeruginosa, a facultative pathogen of red palm weevil, Rhynchophorus ferrugineus. World J Microb Biot, 1995, 11, 618–620.

29    Mazza, G.; Francardi, V.; Simoni, S.; Benvenuti, C.; Cervo, R.; Faleiro, J.R.; Llácer, E.; Longo, S.; Nannelli, R.; Tarasco, E.; Roversi, P.F. An overview on the natural enemies of Rhynchophorus palm weevils, with focus on R. ferrugineus. Biol Control. 2014, 77, 83-92.

31    Caccia, S.; Di Lelio, I.; La Storia, A.; Marinelli, A.; Varricchio, P.; Franzetti, E.; Banyuls, N.; Tettamanti, G.; Casartelli, M.; Giordana, B.; Ferré, J.; Gigliotti, S.; Ercolini, D.; Pennacchio, F., Midgut microbiota and host immunocompetence underlie Bacillus thuringiensis killing mechanism. Proc. Natl. Acad. Sci U. S. A. 2017, 113, 9486-91.

47    Adams, A. S, Currie, C. R.; Cardoza, Y.; Klepzig, K. D.; Raffa, K.F. Effects of symbiotic bacteria and tree chemistry on the growth and reproduction of bark beetle fungal symbionts. Can J For Res. 2009, 39, 1133–47.

51    Cheng, D.; Guo, Z.; Riegler, M.; Xi, Z.; Liang, G.; Xu, Y. Gut symbiont enhances insecticide resistance in a significant pest, the oriental fruit fly Bactrocera dorsalis (Hendel). Microbiome 2017, 5, 13.

72    Verstraete, W. Microbial ecology and environmental biotechnology. ISME J. 2007, 1, 4-8.

77    Sethi, A.; Delatte, J.; Foil, L.; Husseneder, C. Protozoacidal Trojan-Horse: use of a ligand-lytic peptide for selective destruction of symbiotic protozoa within termite guts. PLoS ONE 2014, 9, e106199

79    Gao, H.; Cui, C.; Wang, L.; Jacobs-Lorena, M.; Wang, S. Mosquito microbiota and implications for disease control. Trends Parasitol, 2020, 36, 98-111.

80    Butera, G.; Ferraro, C.; Colazza, S.; Alonzo, G.; Quatrini, P. The culturable bacterial community of frass produced by larvae of Rhynchophorus ferrugineus Olivier (Coleoptera: Curculionidae) in the Canary island date palm. Lett Appl Microbiol. 2012, 54, 530-6.

Reviewer 2 Report

This review summarizes the current understanding of red palm weevil (RPW) interactions with gut microbiota. The authors clearly outline the importance of normal gut flora on RPW development and metabolism (section 2). They also highlight a mechanism by which the RPW modulates its gut microbiota immunologically (section 3). In all the authors mention about a dozen papers that illustrate the impact and the current state of RPW-microbiome and aim to contextualize it with examples from across insects (and even other animals like humans). Finally, they imply the usefulness of RPW gut bacteria as a means of green pest control, thought their connection is tenuous at best relying largely on only a handful of examples from flies.

I think the ideas here are good, but the execution needs development and iteration. I have a few key questions for you to incorporate/address:

  • Is pesticide resistance a problem in RPW? To which types of pesticides? Do gut microbiota have any relationship to this problem, i.e. symbiont-mediate pesticide resistance? What are current strategies and why are they deficient (apart from the environmental/ecological impact)?
  • Can you expand and connect IPM and MRM in RPW? What is your perspective on the integration of these approaches? Offer some specific insights.
  • What kind of green pesticide do you see as the most promising for RPW control considering its cryptic lifestyle? Do you think turning its native flora against it is a viable option? You mention in lines 189-192 that gut bacteria can synergize with pathogens for poor host outcomes, but there is also substantial evidence that normal flora can mitigate pathogenic infections – perhaps there is a more balanced approach to take here.

Additionally, I provide some broader conceptually and organizational suggestions and then some line by line changes.

Broad recommendations:

  • The use of absolutes in the language of this paper is problematic and unscientific. “Always,” “proved,” and “absolutely” should be reworded to reflect that science is about probabilities and rarely speaks with certainty.
  • Both figures need revising. Figure 1 needs references from which this information is being gleaned. Figure 2 requires substantial improvement. Please provide labels for all the structures in this diagram. It’s not clear what is happening to peptidoglycan and why, particularly if peptidoglycan is being monomerized what is that a signal for? It appears that PGRP-LB and PGRP-S1 are inhibiting bacteria in the gut lumen, but it is unclear what the arrow labelled “gut bacterial homeostasis” is referring to and why it doesn’t include these two signals. Additionally, I have no idea what the part containing the antimicrobial peptides is representing. Is that an epithelial cell? In the text, PGRP-L1 & PGRP-L2 are presented as different from PGRP-LB and PGRP-S1, but best I can tell both are negative regulators of the gut bacteria. I think it could be important to also include if these gut bacteria are normal flora, potential pathogens, or both.
  • Given the aim (and title) of this paper, far too little about the author's perspective and recommendations about RPW-specific green pesticides is outlined. Importantly, there appear to be several papers cited in the paper which could provide insight. I think a more in depth discussion of papers 7-10 should be expanded. In fact, the context in which they’re cited does not even suggest that they have much to do with gut microbiota at all. Papers 9 & 10 are specifically addressing something pertinent to this paper – biocontrol potential! In the same vein, section 4 is lackluster. I think the authors need to be more creative in their presentation of their ideas and the diversity of the examples they reference. It’s hard for me to believe that this system is ripe for this kind of endeavor based on the way this is currently structured.

Line-number recommendations

Lines 11-12 & 19-20 are too similar, revise.

Line 15: remove “solidly proved”

Lines 50-51: controlling and detection? Seems like detection is a big issue here too

Lines 65-66: missing something here. there have what?

Lines 77-79: Clarify. Are you saying that dsRNA treatments result in overgrowth of gut bacteria? Are these normal flora or pathogens?

Line 83: What is DUOX? Define.

Line 94-95: remove “absolutely”

Line 97: “suggest”

Line 98: suggested revision, “gut bacteria have to promote growth and development.”

Lines 102-103: suggested revision, “these processes are still unknown.”

Line 125: suggested revision “Previous investigations have identified several PGRPs…”

Line 150: You say many but you only cite 2… this is huge field and this does not even come close to touching the breadth and depth of it.

Line 153: spell out Serratia

Line 161: this is the first time you mention vectors at all

Line 164: suggested revision, “to some extent,”

Line 170: MRM – expand on this idea!

Lines 178-179: There are dozens of examples of this, several much more closely related to your RPW than these fly examples. Expand

Lines 187-188: suggested revision, “This progress strongly suggests….”

Lines 192-194: This is simply not a strong enough place to end. I want to see recommendations, specifics, and measurable deliverables.

Author Response

Dear Reviewer,

Thank you very much for your letter and advice on our manuscript. We have replied to your comments one by one, and revised and adjusted the content of the manuscript under the revision state according to the opinions. Specific replies are as follows:

1: Is pesticide resistance a problem in RPW? To which types of pesticides? Do gut microbiota have any relationship to this problem, i.e. symbiont-mediate pesticide resistance? What are current strategies and why are they deficient (apart from the environmental/ecological impact)?

Response: Done. Yes, pesticide resistance is a problem in RPW. To address it, we have done the following revision: Currently, the exclusionary quarantines, pheromone traps, and bio-control agents are employed to control RPW. However, insecticide applications are still the most effective way for protecting palms from attack by palm weevils [12, 16, 17]. For example, these insecticides, containing organophosphates, carbamates, neonicotinoids, and phenylpyrazoles, have been sprayed onto foliage, used as crown or soil drenches, dressed wound, or injected into trunks or soil at the base of the trunk [12, 18, 19]. Unfortunately, the resistance to the synthetic insecticides, such as chlorpyrifos, imidacloprid and lambda-cyhalothrin, has been determined in the field populations of RPW [20]. Compared with the usage of chemical pesticides, biocontrol have the following advantages, such as low toxicity, target specificity, and sustainability [21]. So the employment of natural enemies is the important alternative to manage the notorious insect pests [9]. Two species of entomopathogenic pathogens, Metarhizium anisopliae [22] and Beauveria bassiana [23, 24], the entomopathogenic nematodes Steinernema carpocapsae [25, 26], and several bacterial species such as Pseudomonas aeruginosa, Serratia marcescens, and Bacillus thuringiensis (Bt) have been shown to be lethal to RPW insects [9, 27]. For instance, lab bioassays revealed that RPW larvae exhibited lower boring activity, and a gradual decrease in feeding behavior upon the challenge of Bt [9]. However, field trials on these biological agents performed so far have showed limited efficacy [12, 28, 29]. Therefore, new association biological control can be exploited for the development of novel management strategies [12]. Recently, it has been uncovered that the pathogenic fungus Beauveria bassiana can interact with the gut microbiota to promote the death of mosquito [30]. The infection of Bt disrupted the gut integrity of Spodoptera littoralis, and then caused the translocation of gut bacteria, especially Serratia and Clostridium, into hemocoel to accelerate the death of this pest [31]. Consequently, these data suggests that the associated microbes of insects have the great potential to drive the development of novel microbes-based pest management strategies [16, 32] (lines 61-86).

2:Can you expand and connect IPM and MRM in RPW? What is your perspective on the integration of these approaches? Offer some specific insights.

Response: Done. We think MRM is a type of IPM. We have expanded MRM as follows: Giving to the profound effects of the associated microbes on insect hosts, Microbial Resource Management (MRM), manipulating and exploiting the microbiota for the management of insect-related problems, has been proposed [51, 72] (lines 191-193). Sterile insect technology (SIT) has been verified as the effective way to decrease the field population size of some insect pests by making and releasing sterile male individuals [35]. However, the mating competitiveness of the sterilized Ceratitis capitate males, made by gama irradiation, was significantly impaired [73]. Interestingly, it was found that the al-terations in gut microbiota are the dominant reason for the impaired mating competitive-ness of sterilized C. captate males. The introduction of gut bacterium, Klebsiella oxytoca, could rescue their mating competitiveness [74]. So this example suggests that it is feasible to solve the insect pest-associated problems by the manipulation of their gut bacteria (lines 193-201).

On the other side, it is much easier to complete the genetic modifications in gut bacteria in contrast with insects. Consequently, genetic engineering has been employed to modify the symbiotic microbes to produce antipathogen effectors (termed paratransgenesis). That is, the dominant gut bacterial species can be modified, by genetic engineering, as the vector to produce some effective molecules to impair the physiology of insect pests [28, 38]. For example, two symbiotic bacteria, Pantoea agglomerans and Serrratia bacterium strain AS1, have been successfully engineered by genetic modifications to secrete five different antimalarial peptides to disrupt the development of Plasmodium falciparum in the guts of Anopheles mosquitoes [75, 76]. Moreover, the termites can be killed by the introduction of the genetically engineered yeast with the capacity to produce a protozoacidal lytic peptide to cause the death of protozoa in their gut [77]. Recently, gut bacteria have been well demonstrated to show the synergistic role to accelerate the mortality of insect pests, which was caused by the pathogenic fungi [30], bacterium [78] and the ingestion of dsRNA [50]. These reports suggest that it is promising to deal with the insect pests-related problems by the manipulation and exploitation of their associated microbes. A central question for the usage of gut bacteria to reduce the infestation of insect pests is how to introduce the bacteria, and to ensure their persistence in their field populations [79]. In RPW larva, two dominant gut bacterial species, Lactococcus lacti and Enterobacter cloacae, play the major roles in the regulation of nutrition metabolism [41]. Interestingly, these two bacterial species have also been found in RPW larval frass and palm trunk tissue [80]. And then the genetically engineered bacteria can be introduced into the RPW larvae through the bacterial inoculation into the healthy palm trunk. So this evidence indicates that L. lacti and E. cloacae are promising candidates for paratransgenesis in this pest. The understanding on the diversity and functionality of intestinal microbiome is the important foundation for the exploitation in MRM approach. Therefore, the deep dissection on the molecular mechanisms underlying the crosstalk between gut bacteria and their insect hosts will accelerate the development of the gut bacteria-based pest management strategies (lines 202-229).

3:What kind of green pesticide do you see as the most promising for RPW control considering its cryptic lifestyle? Do you think turning its native flora against it is a viable option? You mention in lines 189-192 that gut bacteria can synergize with pathogens for poor host outcomes, but there is also substantial evidence that normal flora can mitigate pathogenic infections-perhaps there is a more balanced approach to take here.

Response: Done. The resistance to the synthetic insecticides, such as chlorpyrifos, imidacloprid and lambda-cyhalothrin, has been determined in the field populations of RPW [20] (lines 67-68).

Gut bacteria also mediate the resistance of their hosts to various pathogenic bacteria, parasites and other attacks [28]. Accumulating evidence showed that the axenic insects are more susceptible to pathogens and parasites in contrast with the conventionally reared ones [70, 71]. Interestingly, the germfree RPW larvae exhibited the impaired PO activity and clearing ability to remove the invaded bacteria in hemocoel and died at a significantly higher rate upon the challenge of Serratia marcescens. Moreover, the transcript abundance of several pattern recognition receptors (PRRs) and antimicrobial peptides was significantly decreased in RPW germfree larvae [33]. (lines 172-179). However, it has been determined that the disruption of gut microbiome homeostasis can accelerate host’s death. For example, the exposure of Apis mellifera to glyphosate and antibiotics can perturb its gut bacterial community structure, and then elevate their mortality upon the challenge of the opportunistic pathogen [58, 59] (lines 128-130). Moreover, the disruption of gut microbiome homeostasis leads to the translocation of some bacterial species into hemocoel to become conditioned pathogens and then accelerate host’s death. For example, the infection of Bt disrupted the gut integrity of Spodoptera littoralis, and then caused the translocation of gut bacteria, especially Serratia and Clostridium, into hemocoel to accelerate the death of this pest [31] (lines 82-84).Thus, we think turning its native flora against it is a viable option.

4:The use of absolutes in the language of this paper is problematic and unscientific. “Always,” “proved,” and “absolutely” should be reworded to reflect that science is about probabilities and rarely speaks with certainty.

Response: Done. We have deleted these words in the revised version.

5:Both figures need revising. Figure 1 needs references from which this information is being gleaned. Figure 2 requires substantial improvement. Please provide labels for all the structures in this diagram. It’s not clear what is happening to peptidoglycan and why, particularly if peptidoglycan is being monomerized what is that a signal for? It appears that PGRP-LB and PGRP-S1 are inhibiting bacteria in the gut lumen, but it is unclear what the arrow labelled “gut bacterial homeostasis” is referring to and why it doesn’t include these two signals. Additionally, I have no idea what the part containing the antimicrobial peptides is representing. Is that an epithelial cell? In the text, PGRP-L1 & PGRP-L2 are presented as different from PGRP-LB and PGRP-S1, but best I can tell both are negative regulators of the gut bacteria. I think it could be important to also include if these gut bacteria are normal flora, potential pathogens, or both.

Response: Done. Figure 1: We have revised this figure based on this comment.

Figure 1. Gut bacteria promote the growth and development of RPW larvae via the modulation of nutrient metabolism. (A) The conventionally reared (CR) larvae have higher content of protein, glucose and triglyceride in the hemolymph; (B) the body weight, the content of protein, glucose and triglyceride in the hemolymph of germfree (GF) larvae were significantly decreased [41, 42] (lines 135-139).

Figure 2: This figure was revised according to this comment. The mechanism of PGRP-L1 and PGRP-L2 activate the antimicrobial peptides is still unknown, and need further study.

Figure 2. The role of RPW gut immunity in the regulation of intestinal bacterial homeostasis. Several PGRPs in RPW have confirmed to be involved in the regulation of gut bacterial homeostasis via the different ways [61, 65-67]. PGN: Peptidoglycan. AMP: Antimicrobial peptides. â‘ : Peptidoglycan degradation. â‘¡: The activation of AMP synthesis (lines 167-171).

6:Given the aim (and title) of this paper, far too little about the author's perspective and recommendations about RPW-specific green pesticides is outlined. Importantly, there appear to be several papers cited in the paper which could provide insight. I think a more in depth discussion of papers 7-10 should be expanded. In fact, the context in which they’re cited does not even suggest that they have much to do with gut microbiota at all. Papers 9 & 10 are specifically addressing something pertinent to this paper - biocontrol potential! In the same vein, section 4 is lackluster. I think the authors need to be more creative in their presentation of their ideas and the diversity of the examples they reference. It’s hard for me to believe that this system is ripe for this kind of endeavor based on the way this is currently structured.

Response: Done. We have expended the discussion of paper 9 as follows: So the employment of natural enemies is the important alternative to manage the problems on the notorious insect pests [9] (lines 70-72). And several bacterial species such as Pseudomonas aeruginosa, Serratia marcescens, and Bacillus thuringiensis (Bt) have been shown to be lethal to RPW insects [9, 27]. For instance, lab bioassays revealed that RPW larvae exhibited lower boring activity, and a gradual decrease in feeding behavior upon the challenge of Bt [9] (lines 74-77).

We have added the example as follows: the termites can be killed by the introduction of the genetically engineered yeast with the capacity to produce a protozoacidal lytic peptide to cause the death of protozoa in their gut [77] (lines 210-212).

We have presented our views on the potential of gut bacteria to control pests, and put forward our idea to control RPW by use of gut bacteria, as follows: In RPW larva, two dominant gut bacterial species, Lactococcus lacti and Enterobacter cloacae, play the major roles in the regulation of nutrition metabolism [41]. Interestingly, these two bacterial species have also been found in RPW larval frass and palm trunk tissue [80]. And then the genetically engineered bacteria can be introduced into the RPW larvae through the bacterial inoculation into the healthy palm trunk. So this evidence indicates that L. lacti and E. cloacae are promising candidates for paratransgenesis in this pest. The understanding on the diversity and functionality of intestinal microbiome is the important foundation for the exploitation in MRM approach. Therefore, the deep dissection on the molecular mechanisms underlying the crosstalk between gut bacteria and their insect hosts will accelerate the development of the gut bacteria-based pest management strategies (lines 219-229).

7:Lines 11-12 & 19-20 are too similar, revise.

Response: Done. We have done the following revision: Red palm weevil (RPW), Rynchophorus ferrugineus Olivier, is a destructive pest that often seriously infests the palm plants (lines 11-12).

Red Palm Weevil (RPW), Rhynchophorus ferrugineus Olivier, is a notorious pest, which infests palm trees and has caused great economic losses worldwide (lines 20-21).

8:Line 15: remove “solidly proved”

Response: Done. We have removed it in line 15.

9:Lines 50-51: controlling and detection? Seems like detection is a big issue here too.

Response: Done. We have deleted this sentence in the revised version.

10:Lines 65-66: missing something here. there have what?

Response: Done. We have removed this sentence and done the following revision: Increasing evidence has found that insects have established mutual relationships with their intestinal bacteria [38]. For example … (lines 94-95).

11:Lines 77-79: Clarify. Are you saying that dsRNA treatments result in overgrowth of gut bacteria? Are these normal flora or pathogens?

Response: Done. The dsRNA treatments result in overgrowth of normal flora and alter the structure of gut bacteria. We have done the following revision: Recently, the ingestion of dsRNAs by the willow leaf beetle, Plagiodera versicolora, results in alteration of the commensal gut bacterial structure to accelerate its death [60] (lines 130-132).

12:Line 83: What is DUOX? Define.

Response: Done. We have removed this example in the revised version.

13:Line 94-95: remove “absolutely”

Response: Done. We have done the following revision: When the gut bacteria of RPW larvae were fully removed (line 116).

14:Line 97: “suggest”

Response: Done. We have done the following revision: These data suggest that… (line 119).

15:Line 98: suggested revision, “gut bacteria have to promote growth and development.”

Response: Done. We have done the following revision: gut bacteria can promote the growth and development of RPW larvae (line 120).

16:Lines 102-103: suggested revision, “these processes are still unknown.”

Response: Done. We have done the following revision: However, the molecular mechanisms underlying these processes are still unknown (lines 123-124).

17:Line 125: suggested revision “Previous investigations have identified several PGRPs…”

Response: Done. We have done the following revision: Previous investigations have identified several PGRPs in RPW (lines 151-152),

18:Line 150: You say many but you only cite 2… this is huge field and this does not even come close to touching the breadth and depth of it.

Response: Because the effect of intestinal bacteria on the immune activity of RPW still needs further study, we only cite 2. In addition, we have revised the description as follows: Accumulating evidence showed that the axenic insects are more susceptible to pathogens and parasites in contrast with the conventionally reared ones [70, 71] (lines 173-175).

19:Line 153: spell out Serratia.

Response: Done. We have done the following revision: upon the challenge of Serratia marcescens (line 177).

20:Line 161: this is the first time you mention vectors at all.

Response: Because it is much easier to realize the genetic modification in gut bacteria in contrast with insects, the dominant gut bacterial species can be modified, by genetic engineering, as the vector to produce some effective molecules to impair the physiology of insect pests. We mentioned it in the last paragraph to make a point about RPW control.

21:Line 164: suggested revision, “to some extent,”

Response: Done. We have removed this sentence.

22:Line 170: MRM – expand on this idea!

Response: Done. We have expand the MRM as follows: Giving to the profound effects of the associated microbes on insect hosts, Microbial Resource Management (MRM), manipulating and exploiting the microbiota for the management of insect-related problems, has been proposed [51, 72] (lines 191-193).

23:Lines 178-179: There are dozens of examples of this, several much more closely related to your RPW than these fly examples. Expand

Response: Done. We have not expanded the examples. The example of sterile insect technology in Ceratitis capitat and the effect of gut bacteria on its mating competitiveness is classical, so we cited it to illustrate that it is feasible to solve the insect pest-associated problems by the manipulation of their gut bacteria.

24:Lines 187-188: suggested revision, “This progress strongly suggests….”

Response: Done. We have done the following revision: These reports suggest that it is promising to deal with the insect pests-related problems by the manipulation and exploitation of their associated microbes (lines 215-216).

25:Lines 192-194: This is simply not a strong enough place to end. I want to see recommendations, specifics, and measurable deliverables.

Response: Done. We have done the following revision: A central question for the usage of gut bacteria to reduce the infestation of insect pests is how to introduce the bacteria, and to ensure their persistence in their field populations [79]. In RPW larva, two dominant gut bacterial species, Lactococcus lacti and Enterobacter cloacae, play the major roles in the regulation of nutrition metabolism [41]. Interestingly, these two bacterial species have also been found in RPW larval frass and palm trunk tissue [80]. And then the genetically engineered bacteria can be introduced into the RPW larvae through the bacterial inoculation into the healthy palm trunk. So this evidence indicates that L. lacti and E. cloacae are promising candidates for paratransgenesis in this pest. The understanding on the diversity and functionality of intestinal microbiome is the important foundation for the exploitation in MRM approach. Therefore, the deep dissection on the molecular mechanisms underlying the crosstalk between gut bacteria and their insect hosts will accelerate the development of the gut bacteria-based pest management strategies (lines 217-229).

Round 2

Reviewer 1 Report

The authors have carefully addressed all of my original comments and those of other reviewers. I have no further suggestions for improvement. 

Reviewer 2 Report

It appears that the authors have taken the time to revise and improve their manuscript. The result is both presents a more compelling narrative and communicates their ideas more directly.